# Ascorbate Plus Buformin in AML: A Metabolic Targeted Treatment

**DOI:** 10.3390/cancers14102565

**Published:** 2022-05-23

**Authors:** Cristina Banella, Gianfranco Catalano, Serena Travaglini, Elvira Pelosi, Tiziana Ottone, Alessandra Zaza, Gisella Guerrera, Daniela Francesca Angelini, Pasquale Niscola, Mariadomenica Divona, Luca Battistini, Maria Screnci, Emanuele Ammatuna, Ugo Testa, Clara Nervi, Maria Teresa Voso, Nelida Ines Noguera

**Affiliations:** 1Neurooncoemtology Units, Santa Lucia Foundation, I.R.C.C.S., 00143 Rome, Italy; cristina.banella@meyer.it (C.B.); gianfranco.catalano@uniroma2.it (G.C.); serenatravaglini@live.it (S.T.); tiziana.ottone@uniroma2.it (T.O.); zazaalessandra96@gmail.com (A.Z.); 2Department of Health Sciences, Meyer Children’s University Hospital, 50139 Florence, Italy; 3Department of Biomedicine and Prevention, University of Rome Tor Vergata, 00133 Rome, Italy; 4Department of Hematology, Oncology and Molecular Medicine, Istituto Superiore di Sanità, 00161 Rome, Italy; elvira.pelosi@iss.it (E.P.); ugo.testa@iss.it (U.T.); 5Neuroimmunology and Flow Cytometry Units, Santa Lucia Foundation, I.R.C.C.S., 00143 Rome, Italy; g.guerrera@hsantalucia.it (G.G.); df.angelini@hsantalucia.it (D.F.A.); l.battistini@hsantalucia.it (L.B.); 6Hematology Unit, Saint’ Eugenio Hospital, University of Rome Tor Vergata, 00144 Rome, Italy; pniscola@aslrmc.it; 7Policlinico Tor Vergata, University of Rome Tor Vergata, 00133 Rome, Italy; mariadomenica.divona@ptvonline.it; 8Banca Regionale Sangue Cordone Ombelicale UOC Immunoematologia e Medicina Trasfusionale, Policlinico Umberto I, 00161 Roma, Italy; m.screnci@policlinicoumberto1.it; 9Department of Hematology, University Medical Center Groningen, 9713 GZ Groningen, The Netherlands; e.ammatuna@umcg.nl; 10Department of Medical and Surgical Sciences and Biotechnologies, University of Roma La Sapienza, 04100 Latina, Italy; clara.nervi@uniroma1.it

**Keywords:** Acute Myeloid Leukemia, Seahorse XF, metabolism, pharmacologic activity, ascorbate, buformin, OXPHOS, glycolysis, hexokinase 1/2, GLUT1

## Abstract

**Simple Summary:**

Acute Myeloid Leukemias (AMLs) are rapidly progressive clonal neoplastic diseases. The overall 5-year survival rate is very poor: less than 5% in older patients aged over 65 years old. Elderly AML patients are often “unfit” for intensive chemotherapy, further highlighting the need of highly effective, well-tolerated new treatment options for AMLs. Growing evidence indicates that AML blasts feature a highly diverse and flexible metabolism consistent with the aggressiveness of the disease. Based on these evidences, we targeted the metabolic peculiarity and plasticity of AML cells with an association of ascorbate, which causes oxidative stress and interferes with hexokinase activity, and buformin, which completely shuts down mitochondrial contributions in ATP production. The ascorbate–buformin combination could be an innovative therapeutic option for elderly AML patients that are resistant to therapy.

**Abstract:**

In the present study, we characterized the metabolic background of different Acute Myeloid Leukemias’ (AMLs) cells and described a heterogeneous and highly flexible energetic metabolism. Using the Seahorse XF Agilent, we compared the metabolism of normal hematopoietic progenitors with that of primary AML blasts and five different AML cell lines. We assessed the efficacy and mechanism of action of the association of high doses of ascorbate, a powerful oxidant, with the metabolic inhibitor buformin, which inhibits mitochondrial complex I and completely shuts down mitochondrial contributions in ATP production. Primary blasts from seventeen AML patients, assayed for annexin V and live/dead exclusion by flow cytometry, showed an increase in the apoptotic effect using the drug combination, as compared with ascorbate alone. We show that ascorbate inhibits glycolysis through interfering with HK1/2 and GLUT1 functions in hematopoietic cells. Ascorbate combined with buformin decreases mitochondrial respiration and ATP production and downregulates glycolysis, enhancing the apoptotic effect of ascorbate in primary blasts from AMLs and sparing normal CD34+ bone marrow progenitors. In conclusion, our data have therapeutic implications especially in fragile patients since both agents have an excellent safety profile, and the data also support the clinical evaluation of ascorbate–buformin in association with different mechanism drugs for the treatment of refractory/relapsing AML patients with no other therapeutic options.

## 1. Introduction

Acute Myeloid Leukemias (AMLs) are rapidly progressive clonal neoplastic diseases, which derive from hematologic stem cells that have lost their homeostatic capacity [1,2]. AMLs are primarily diseases of older adults, with a median age of approximately 70 years old at diagnosis [3]. The incidence increases from 2–3 per 100,000 in young adults to 13 to 15 per 100,000 in the seventh and eighth decades of life. The overall 5-year survival rate is very poor: less than 5% in older patients aged over 65 years old [4]. Therapeutic efforts are often frustrated by AML clonal evolution and resistance to treatments, arising even in younger patients. Elderly AML patients are often “unfit” for intensive chemotherapy, further highlighting the need of highly effective, well-tolerated new treatment options for AMLs.

Growing evidence indicates that AML blasts feature a highly diverse and flexible metabolism consistent with the aggressiveness of the disease [5,6,7]. Aberrant enzymatic activity cooperates with mutations of tumor suppressors and oncogenes in disease progression. For example, AML cells reduce both host insulin sensitivity and secretion to increase glucose availability for malignant cells [8]. The glycolytic pathway sustains leukemia maintenance and progression. The AML bulk, stem cells and their progeny had a greater mitochondrial mass and higher rates of oxygen consumption compared to a normal hematopoietic progenitor [9]. Moreover, leukemia stem cells (LSCs) are characterized by low rates of energy metabolism and a low cellular oxidative status (termed “ROS-low”). Surprisingly, ROS-low cells are unable to utilize glycolysis when mitochondrial respiration is inhibited. Thus, the maintenance of mitochondrial function is essential for LSC survival [10]. The mitochondrial oxidative phosphorylation system (OXPHOS) is sustained by elevated amino acid metabolism in LSCs from AML [11,12]. Resistant LSCs exhibit higher mitochondrial oxygen consumption that is dependent on increased tricarboxylic acid (TCA) cycle activity and fatty acid oxidation (FAO). Importantly, at diagnosis, high and low OXPHOS AML cells coexist, while after chemotherapy, high OXPHOS cells predominantly persist and survive [13]. Clonal heterogeneity and metabolic heterogeneity are, in general, associated with the failure of anti-cancer drugs, including metabolic inhibitors [14,15,16]. However, the metabolic reprogramming occurring in AML hematopoietic stem cells depends on their genetic characteristics, and thus represents a promising target for treatment [17,18,19,20,21]. In this study, we compared the metabolic pathways underlying the different stages of differentiation of normal myeloid progenitors/precursors with that of primary blasts from six AML patients and of five AML cell lines featuring different genetic mutations commonly associated with AMLs, and diverse metabolic phenotypes. We found that AML cells have a reduced respiratory capacity and reduced glycolytic reserve than their normal cellular counterpart. Therefore, we hypothesized that resistance to therapy can be overcome by associating drugs with complemental mechanisms of action and targeting specific metabolic features of AML cells.

Ascorbic acid (vitamin C) at pharmacological concentrations has pro-oxidant [22,23,24] and anti-cancer activities, as reported by us and others [25,26,27]. Ascorbate inhibits hexokinase activity [28], which produces glucose-6-phosphate to initiate two major metabolic pathways: glycolysis and the pentose phosphate pathway. Hexokinase 1 and 2 (HK1/HK2) are also associated with the mitochondrial membrane permeability transition pore (PTP) and prevents apoptosis, thus controlling reactive oxygen species (ROS) formation [29]. Recent studies reported the capacity of ascorbate to target leukemia-initiating cells [20,21].

The oral biguanide metformin is an anti-diabetic drug that delays the gastrointestinal absorption of glucose, increases insulin sensitivity and intracellular glucose uptake and inhibits liver glucose synthesis. It induces apoptosis and inhibits tumor growth in vitro and in vivo in malignancies, including breast cancer [30,31,32,33], lung cancer [34,35], melanoma [36,37] and hepatocellular cancer [38]. Buformin (1-butylbiguanide) is an analog of metformin and is a more potent inhibitor of the mitochondrial complex I of the electron transporter chain and abolishes mitochondrial respiration. The drug inhibits tumor growth in endometrial uterine cancer [39] and is currently being tested in a trial for the treatment of diffuse large B-cell lymphoma (ClinicalTrials.gov Identifier NCT02871869).

Based on these evidences, we targeted the metabolic peculiarity and plasticity of AML cells with an association of ascorbate to induce an oxidative stress and to interfere with hexokinase activity [28] and buformin to shut down the mitochondrial contribution in ATP production. Our data support the clinical evaluation of the ascorbate–buformin combination as a treatment option for refractory/relapsing AML patients and for older and unfit patients.

## 2. Materials and Methods

### 2.1. Primary Patient Samples and Controls

Bone marrow (BM) samples were collected from 17 consecutive newly diagnosed de novo AML patients admitted at the Department of Hematology of the University of Rome Tor Vergata. All samples had at least a 70% infiltration by leukemic blasts. Normal bone marrow (NBM) and CD34+ hematopoietic progenitors, isolated from the cord blood (CB) of healthy full-term placentas, were used as controls. Written informed consent was obtained from all patients in accordance with the Declaration of Helsinki and the study was approved by the ethical committee of the University of Rome Tor Vergata. The CD34+ cells were purified from the CB and BM by positive selection using the midiMACS immunomagnetic separation system (Miltenyi Biotec, Bergisch Gladabach, Germany), according to the manufacturer’s instructions. The purity of the CD34+ cells was assessed by flow cytometry using a monoclonal PE-conjugated anti-CD34 antibody and resulted in a purity of over 95% (range 92–98%). Purified human hematopoietic progenitor cells were grown in a serum-free medium: serum substitute BIT 9500 (Stem cell Technologies, Vancouver, BC, Canada). The CD34+ cells were induced into a promyelocyte (Day 7) and granulocytic differentiation (Day 13) by the addition of IL-3 (1 unit/mL), GM-CSF (0.1 ng/mL) and saturating amounts of G-CSF (500 units/mL). The morphologic and immunophenotype characterization of the cells grown under these conditions has been previously described in detail by our group [40]. Cytogenetic and molecular analyses of the AML blasts were performed as described [41] and are reported in Table 1.

Normal bone marrow (NBM) cells and hematopoietic CD34+ progenitor cells that were isolated from the cord blood (CB) obtained from healthy full-term placentas were used as the controls.

### 2.2. Cell Lines and Cell Culture

U937-Mock cells (the U937 monoblastic cell line transfected with the empty Zn-inducible MT1 promoter vector) were used as the controls; U937-AETO (a zinc-inducible RUNX1/RUNX1T1 model), OCI-AML3 (an AML-M4-derived cell line carrying an NPM1 gene mutation (type A) and the DNMT3A R882C mutation) and OCI-AML2 (an AML-M4-derived cell line carrying the DNMT3A R635W mutation) were kindly provided by Emanuela Colombo, European Institute of Oncology, Milan, Italy. MV4-11 (a biphenotypic B myelomonocytic leukemia carrying the FLT3-ITD mutation and MLL/AF4 translocation) was purchased from the Leibniz Institute DSMZ-German Collection of Microorganisms and Cell Cultures (Braunschweig, Germany). The cells were cultured in an RPMI medium (Euroclone; Pero, MI, Italy), 10% fetal bovine serum (FBS) (GIBCO-BRL), 20 mM of Hepes, 100 U/mL of penicillin and 100 µg/mL of streptomycin (GIBCO-BRL). The cultures were maintained at 37 °C in a 5% CO_2_ humidified incubator.

### 2.3. Cell Viability

A CellTiter 96^®^ AQueous One Solution Cell Proliferation Assay was used. The AML cells were seeded in a 96-well plate at an initial density of 1 × 10^4^ cell/well and were treated with 1 mM of ascorbate; 0.1 mM of buformin; or both for 72 h at 37 °C. Subsequently, 5 μL of the CellTiter 96^®^ AQueous One Solution Cell Proliferation Assay (Promega; Madison, WI, USA) were added to each well and the cells were incubated for 4 h. The absorbance was read at 490 nm using a microplate reader (Thermo Scientific™ Varioskan™ Flash Multimode Reader; Waltham, MA, USA). The cell viability was assessed by comparison with the control cells treated with the vehicle alone. At least 3 independent biological replicates were performed.

A CellTiter-Glo^®^ Luminescent Cell Viability Assay is a homogeneous method for determining the number of viable cells in a culture. It is based on the quantitation of ATP, an indicator of metabolically active cells. Briefly, the cells were plated at 2 × 10^4^ cell/mL/well and were cultured for 72 h with or without the treatments reported above. The intracellular ATP levels were determined using the CellTiter-Glo^®^ Substrate Assay System (Promega; Madison, WI, USA) according to the manufacturer’s instructions. At least 3 independent biological replicates were performed.

### 2.4. Colony Formation Unit Assay

For the Colony-Forming Unit (CFU) Assay, the AML cells were seeded at 70,000 cells/mL in a Methocult 4035 medium (STEMCELL Technologies, Tukwila, WA, USA), following the manufacturer instructions, and were incubated at 37 °C at 5% pCO_2_ in a humidified incubator with the addition or not of 0.1 mM of buformin and/or 1 mM of ascorbate. The healthy BM mononucleated cells were cultured as described but were seeded at a density of 20,000 cells/mL, the purified CD34+ cells at 100 cells/mL and the AML cell lines at 300 cells/mL, as recommended by the Methocult medium manufacturer instructions. The number of colonies were counted under a phase-contrast optical microscope after 8 days of culture [42].

### 2.5. Western Blot Analysis

Cell pellets were resuspended in a lysis buffer with 10 mM of Tris-HCl (pH 7.4), 5 mM of EDTA, 150 mM of NaCl, 1% Triton X-100, 250 μM of orthovanadate, 20 mM of β-glycerophosphate and protease inhibitors (Sigma-Aldrich, Steinheim, Germany). The lysates were centrifuged at 10,000× *g* for 30 min at 4 °C and the supernatants were stored at −80 °C. The protein concentration was measured by the Bradford Assay (#500–0006; Bio-Rad, München, Germany). Thirty microgram aliquots of proteins were re-suspended in a reducing Laemmli Buffer (with β-mercaptoethanol), loaded onto a 12% polyacrylamide gel and then transferred to a PVDF membrane. After blocking with 5% milk (Fluka, Sigma-Aldrich, Saint Louis, HI, USA), the membranes were incubated with primary antibodies (Appendix A). Horseradish peroxidase-conjugated IgG preparations were used as secondary antibodies, and immunoreactivity was determined by the enhanced chemiluminescence (ECL) method (Amersham, Buckinghamshire, UK). The autoradiograms were exported for densitometry analysis. The protein signal intensities were measured using the Quantity One Software (Bio-Rad Laboratories, Hercules, CA, USA). The signal quantity was normalized using the unrelated protein β-actin (Cell Signaling Technology, Beverley, MA, USA) [43].

### 2.6. Metabolic Assays

Mitochondrial and glycolytic functions were assessed using a Seahorse Bioscience XFe96 analyzer in combination with the Seahorse Bioscience XF Cell Mito Stress Test and the Bioscience XF Cell Glycolysis Stress Test (Agilent Technologies, Santa Clara, CA, USA), respectively, as described [44]. Briefly, the extracellular acidification rate (ECAR), reflecting the conversion of glucose to lactate and resulting in a net production of protons in the extracellular medium, was measured directly using the Agilent Seahorse XF instrument. The cells were first injected with saturating concentrations of glucose (10 mM). The glucose-induced response is reported as the rate of glycolysis under basal conditions. The second injection was 2 µM of oligomycin, an ATP synthase inhibitor. Oligomycin inhibits mitochondrial ATP production and shifts the energy production to glycolysis, with the subsequent increase in ECAR revealing the maximum cellular glycolytic capacity. The final injection was 50 mM of 2-deoxy-glucose (2-DG), a glucose analog that inhibits glycolysis, confirming that the ECAR produced in the experiment was due to glycolysis. The difference between the glycolytic capacity and basal glycolysis defines the glycolytic reserve.

Mitochondrial respiration is directly measured by the oxygen consumption rate (OCR) of cells. Basal respiration represents the energetic demand of the cell under baseline conditions. Oligomycin is first injected in the assay and results in a reduction in the mitochondrial respiration or OCR. The decrease in the OCR is linked to cellular ATP production. Carbonyl cyanide-4 (trifluoromethoxy) phenylhydrazone (FCCP) is an uncoupling agent and is injected at concentrations of 1 µM following oligomycin. As a result, the electron flow through the ETC is uninhibited, and oxygen consumption reaches the maximum. This is the maximum respiration. The spare respiratory capacity, defined as the difference between maximal and basal respiration, is a measure of the ability of the cell to respond to an increased energy demand. A proton leak has its origin in the fact that no living cell converts all the energy of the proton gradient to ATP, meaning oxidative phosphorylation is incompletely ‘coupled’ since protons can ‘leak’ across the inner membrane and thus balance the gradient without ATP synthesis.

The percentage of ATP production from glycolysis and mitochondrial respiration was measured using the XF Real-Time ATP Rate Assay (Agilent Technologies). This test uses metabolic modulators (oligomycin and a mix of rotenone and antimycin A, that when serially injected, allows the calculation of the mitochondrial and glycolytic ATP production rates).

The rate of oxidation of each fuel (pyruvate, fatty acids and glutamine) was determined using the XF Mito Fuel Flex Test (Agilent Technologies).

The cells’ mitochondrial dependency and flexibility for the usage of each of the fuel sources were determined by measuring the decrease in fuel oxidation (the decline in the OCR) upon addition of one or more inhibitors, including UK5099, which blocks the glucose oxidation pathway, BPTES, which blocks the glutamine oxidation pathway and Etomoxir, an inhibitor of long-chain fatty acid oxidation. Sequentially inhibiting the pathway of interest enables the calculation of how dependent the cells are on the pathway to meet the basal energy demand. Inhibiting the two alternative pathways enables the calculation of the cells’ mitochondrial capacity to meet energy demands using another fuel. Fuel flexibility is calculated by subtracting the fuel dependency from the fuel capacity.

### 2.7. Cytofluorimetric Analysis

The markings were performed using 0.5 × 10^6^ cells resuspended in a volume of 100 µL of current buffer (PBS + 1% FBS + 0.5% EDTA 500 mM). Cells were labeled with annexin V, which binds to the phosphatidylserine (PS) externalized on the surface of cell membranes and is used to measure early stage apoptosis, and ‘Live Dead’, which enters cells with damaged membranes and is used to assess the terminal stages of apoptosis. After a 15 min incubation in the dark at room temperature, the cells were washed and resuspended in 100 µL binding buffer 1×. The analysis of the samples was performed using the CytoFLEX flow cytometer (Beckman Coulter, Brea, CA, USA) equipped with three lasers. About 500,000 cells were selected for each sample based on physical size (FSC) and graininess parameters. The cells labeled with a single fluorochrome were used as controls to adjust the compensation. The data were compensated and analyzed using the FlowJo software (TreeStar, Ashland, OR, USA) [45].

### 2.8. Statistical Analysis

Data were analyzed using GraphPad Prism 6 (GraphPad Software Inc., San Diego, CA, USA). Statistical analyses were performed using the Student’s *t*-test, Mann–Whitney test, Kruskal–Wallis one-way ANOVA and Dunn’s post hoc tests or the one-way ANOVA and Tukey’s multiple comparison test as indicated. Statistical significance was established at *p* < 0.05.

## 3. Results

### 3.1. Metabolic Dependence of Primary AML Blasts

To investigate the whole-substrate oxidation usage of primary AML cells, we evaluated the oxygen consumption rate (OCR) and extracellular acidification rate (ECAR) in primary blasts isolated from six AML patients (numbers 1, 5, 6, 9, 12 and 16 from Table 1) and in five human AML cell lines. The metabolic peculiarities of the leukemic cells were compared to those of normal hematopoietic precursors undergoing different stages of myeloid maturation/differentiation, as indicated by morphological and immunophenotypic changes. Primary AML blasts showed a heterogenous glucose consumption, while basal glycolysis, the glycolytic reserve and the glycolytic capacity were higher with respect to NBM but comparable to values measured in early progenitors/precursors (EP/Ps) from the CB CD34+ cells at day 13 of culture (N13, mostly granulocytic differentiated cells: CD11b 72%, CD15 80% and CD34 7%) (Figure 1a and Table 2). In line with literature reports, AML cells displayed high glucose consumption and heavily relied on it [46]. Interestingly, the EP/Ps at day 7 of culture (N7, mostly promyelocytes: CD11b Pe 16.8%, CD13 Pe 86.6%, CD14 Pe 5.1%, CD15 Pe 3.2% and CD34 Pe 45.2%) showed a higher glycolytic reserve than the AML blasts (*p* = 0.02) (Figure 1a and Table 2). Of note, at day 7 and day 13, the EP/Ps were highly proliferating cells exposed to high concentrations of growth factors, which could have affected their metabolic activities.

Regarding mitochondrial respiration, the AML blasts primary presented lower values in the spare respiratory capacity with respect to the EP/Ps (N7) (*p* = 0.02), indicating that they were more sensitive to oxidative stress than normal progenitors, as previously reported by Sriskanthadevan et al. [47]. This finding indicates that the therapeutic targeting of mitochondrial respiration can be useful in AMLs. The measurements of basal respiration and proton leaks indicated that the AML cells were distributed in two populations, one subset presenting higher values and another with lower values (Figure 1b and Table 3).

### 3.2. Metabolic Dependency in AML Cell Lines

Three AML cell lines were used as metabolic models for the AMLs: OCI-AML2 (carrying the DNMT3A R635W mutation), OCI-AML3 (carrying the NPM1 gene mutation type A and the DNMT3A R882C mutation) and MV4-11 (carrying the FLT3-ITD mutation and MLL/AF4 translocation). The OCI-AML2 and OCI-AML3 cells showed higher glycolysis basal values with respect to MV4-11, the OCI-AML3 cells presented the highest levels of glycolytic capacity and glycolytic reserve (Figure 2a and Table 4) and the MV4-11 cells had greater OXPHOS values compared with the OCI-AML2 and OCI-AML3 cells (Figure 2b and Table 5).

Then, we analyzed the fuel used in mitochondrial respiration. The energy produced by the cells derived from the mitochondrial oxidation of glucose, glutamine and fatty acids. Dependency indicates that the cells’ mitochondria were unable to compensate for the blocked pathway by oxidizing other fuels. Flexibility indicates the cells’ mitochondria had the ability to compensate for the inhibited pathway by using other pathways to fuel mitochondrial respiration. The MV4-11 cells displayed a significant dependency on fatty acid oxidation (FAO) and slightly on glycolysis. OCI-AML3 strongly depended on glucose and slightly on FAs and glutamine. OCI-AML2 depended partially on glycolysis and FAs. OCI-AML3, OCI-AML2 and MV4-11 showed the highest flexibility towards all three fuels (Figure 2c, The uncropped Western blots have been shown in Appendix A). We analyzed the expression levels of two FAO key proteins, Carnitine transporter CT2 (SLC22A16) and Carnitine palmitoyl transferase I (CPT1A), and found that both of them were expressed at higher levels in the MV4-11 cells, which are highly dependent on FAs, with respect to the OCI-AML2 and OCI-AML3 cells (Figure 2d and Table 6).

### 3.3. Action of Buformin on OXPHOS Metabolism in AML Cells

Since buformin targets mitochondrial complex I [39], we evaluated the drug concentration causing 50% OXPHOS inhibition (IC50) in the AML cell lines. We treated the three AML cell lines with 0–10–50–100 µM of buformin for 24 h and evaluated the OCR at baseline and following the addition of: oligomycin-A (an ATP synthase inhibitor), to determine the amount of oxygen consumption coupled to ATP synthesis; FCCP (releases electron flow through the ETC, thus maximizing oxygen consumption) to determine the maximal respiratory capacity; and antimycin A plus rotenone (electron transporter complex III and I inhibitors) to determine the spare respiratory capacity. Our result showed that buformin inhibited the OXPHOS activity in a concentration-dependent manner, with IC50s of 49 µM for OCI-AML3, 63 µM for MV4-11 and 103 µM for OCI-AML2 (Figure 3a). These results indicate that treatment with 100µM of buformin strongly affected mitochondrial ATP production in these AML cell lines.

### 3.4. Effect of Ascorbate on Glycolytic Metabolism in AML Cells

From our previous experience in treating AML cells with ascorbate, we identified the 3 mM concentration as rapidly cytotoxic due to its pro-oxidant effect [25]. To investigate the metabolic effect of the drug, in this study AML cells were treated with 1 mM of ascorbate, which significantly inhibited the expression levels of Hexokinase II (HK2), a main glycolysis-initiating enzyme in the hemopoietic system (Figure 3b) and in tumors [28]. Similar results were obtained using an antibody recognizing both Hexokinase I and HK2 (HK1/2) (Figure 3b). In addition, we investigated the effect of ascorbate on GLUT1, a transporter of glucose in hematopoietic cells [48]. We found that ascorbate significantly decreased GLUT1 expression levels in OCI-AML2 and OCI-AML3 (Figure 3c), further suggesting an effect of ascorbate on glycolysis in AML cells.

To confirm this, the XF Glycolytic Stress Test was used to determine the basal rates of glycolysis by measuring the percentage of increase in the ECAR after the addition of glucose. The glycolytic reserve is a measure of the difference in the ECAR before and after treatment with oligomycin (which inhibits ATP synthase and thereby the OXPHOS’ ATP). The glycolytic reserve reflects the compensatory increase in glycolysis corresponding to the level of ATP no longer produced by the OXPHOS. It was found that 1 mM of ascorbate did not affect glycolysis in the MV4-11 and OCI-AML2 cells, whereas it lowered basal glycolysis by 1.7 folds (*p* = 0.01) and the glycolysis capacity by 1.8 folds in the OCI-AML3 cells (*p* = 0.01) (Figure 3c).

### 3.5. Metabolic Background Influences Apoptotic Response to Metabolic Treatments of AML Cell Lines

We then treated the OCI-AML2, OCI-AML3 and MV4-11 cell lines with ascorbate (1 mM), buformin (0.1 and 0.5 mM) and with an ascorbate–buformin combination for 72 h, measuring apoptosis with cytofluorimetric analysis. The OCI-AML3 cells showed a strong apoptotic response to the combined ascorbate–buformin treatment; buformin potentiated the ascorbate effect on apoptosis at both of the concentrations used. The MV4-11 cells were sensitive to 0.5 mM of buformin and 0.5 mM of ascorbate–buformin; the combination did not increase the apoptotic effect. In the OCI-AML2 cells, only the combination treatment significantly enhanced apoptosis (Figure 3d and Table 7). Of note that the OCI-AML3 cells, which are highly dependent on glycolysis, were the most responsive cells, whereas only modest effects were induced by these treatments in the OCI-AML2 and MV4-11 cells, presenting a lower glycolysis dependence and high fuel flexibility.

### 3.6. Ascorbate Plus Buformin Combination Treatment Effectively Induces Apoptosis in Primary AML Blasts

The treatment of primary blasts from seventeen AML patients (Table 1) with ascorbate (1 mM) or buformin (0.1 and 0.5 mM) as the sole agent for 72 h induced a moderate increase in the percentage of apoptotic cells as detected by cytofluorimetric analysis (ascorbate *p* < 0.05; buformin 0.1 mM *p* < 0.0005). Remarkably, the combined ascorbate–buformin treatment significantly increased the apoptotic rate of the AML blasts. We did not observe an association with genetic alterations, probably due to the low number of samples analyzed (Figure 4a and Table 7).

Next, we studied the effect of the ascorbate, buformin and ascorbate–buformin treatments on hematopoietic colony formation. The AML cell lines treated with the ascorbate or ascorbate–buformin treatments produced a significantly lower number of colonies as compared to the cells treated with buformin or the control cells (Figure 4b and Table 8). Similar results were obtained by treating the AML primary blasts with these agents. (Figure 4c). Interestingly, the same treatments did not affect the clonogenic activity of the BM cells from healthy donors (BMC) or of the CD34+ cells purified from CB, indicating low or no toxicity of the ascorbate–buformin combination treatment on normal hematopoietic progenitors. (Figure 4d and Table 8).

### 3.7. Ascorbate Plus Buformin Inhibits Glycolysis in AML Cells

By performing the XF Glycolytic Stress Test, we observed that the ascorbate treatment decreased basal glycolysis by 1.6 times (*p* < 0.01), buformin by 2.0 times (*p* < 0.001) and the ascorbate–buformin combination by 2.1 times (*p* < 0.001) in the OCI-AML3 cells. In this AML cell line, the glycolytic capacity was reduced by 1.8 times following the ascorbate (*p* < 0.01), 3.5 times following the buformin (*p* < 0.0001) and 3.2 times after the ascorbate–buformin combination treatment (*p* < 0.0001). The glycolytic reserve was completely Erased by the buformin (*p* < 0.0001) and ascorbate–buformin combinatory treatments (*p* < 0.0001) (Figure 5a and Appendix A). These results are in line with the high rate of apoptosis induced by these treatments in the OCI-AML3 cells (Figure 3d). The untreated MV4-11 cells showed a lower basal glycolytic rate as compared with the other cell lines. In these cells, the buformin induced an increase of 1.8 times in basal glycolysis (*p* < 0.01) and the ascorbate–buformin combination treatment of 2.4 times (*p* < 0.0001). The glycolytic capacity was enhanced by 1.7 times (*p* < 0.01) by the ascorbate–buformin treatment (Figure 5a and Appendix A). This event may account, at least in part, for the absence of the potentiation effect on apoptosis by the ascorbate–buformin combination treatment in the MV4-11 cells (Figure 3d). The OCI-AML2 metabolic status was not significantly affected by these treatments (Figure 5a and Appendix A), which is in accordance with their lower apoptotic response (Figure 3d).

In order to confirm our results and to exclude intrinsic cellular variability, we utilized U937-AETO/U937-Mock cell lines (the RUNX1/RUNX1T1 inducible system). In U937-AETO, similar to OCI-AML3, the ascorbate–buformin treatment inhibited basal glycolysis by 1.9 times (*p* < 0.05) and the glycolytic capacity by 2.3 times (*p* < 0.005), with respect to the control U937-Mock cells. The glycolytic reserve was completely erased by the buformin and ascorbate–buformin treatments (*p* < 0.005). This effect did not occur in the U937-Mock control cells (Appendix A). The cellular viability by the MTS assay demonstrated that the RUNX1/RUNX1T1 expression induced a higher sensitivity of the U937 cell to 1 mM of ascorbate (*p* < 0.001) (Table 9 and Appendix A), 0.1 mM of ascorbate–buformin (*p* < 0.0001) and 0.5 mM of ascorbate–buformin (*p* < 0.0001). These results indicated that the RUNX1/RUNX1T1 oncoprotein rendered the cells dependent on glycolytic metabolism, confirming the relationship between the inhibition of glycolysis and sensitivity to the ascorbate and ascorbate–buformin treatment in AMLs.

Interestingly, the primary blasts from an AML patient (N°15, Table 1) with a high dependence on glycolysis in the basal conditions showed, as all the AML cell lines analyzed, OXPHOS inhibition after treatment with buformin or with the ascorbate–buformin combination (Appendix A). To note in these cells, the ascorbate–buformin combination decreased basal glycolysis by two times and the glycolytic capacity by 2.5 times (Figure 5b). Those cells resulted highly sensitive to the induction of apoptosis by the ascorbate and ascorbate–buformin treatments (Figure 5c), further confirming that these treatments are mostly effective in cells that are highly dependent on glycolysis. Important analyzing the six samples for metabolic and apoptotic values we found a positive correlation between apoptosis and glycolytic levels in the AML blast treated with both the ascorbate and ascorbate–buformin treatments (Pearson’s coefficient r = 0.7) (*p* = 0.1) (Figure 5d); even if these results are not significant, which would probably be due to the low number of samples, there is a clear trend in line with our previous results.

### 3.8. Resistance to Ascorbate–Buformin Combined Treatment Depends on Metabolic Plasticity of AML Cells

To characterize the metabolic background of resistance to these treatments, we evaluated the effect of 1 mM of ascorbate, 0.1 mM of buformin and the ascorbate–buformin combination on the metabolic behavior of the OCI-AML2, OCI-AML3 and MV4-11 cell lines. The OXPHOS was inhibited by the buformin and by the ascorbate–buformin combination in all the three cell lines (Figure 3a and Appendix A). As pointed out before, glycolysis was affected differently in the cell lines. The OCI-AML2 cells did not show significant variations, the OCI-AML3 cells showed inhibition and the MV4-11 cells showed increased glycolysis (Figure 5a and Appendix A). This suggests that glycolytic behavior is the discriminant between the response and resistance to the treatment, in line with the higher sensitivity observed in the glycolysis-dependent OCI-AML3 cells. Conversely, the OCI-AML2 cells and the more flexible MV4-11 cells, which did not rely upon glycolysis in the basal conditions, escaped the ascorbic–buformin treatment, probably switching to glycolytic metabolism. Since these treatments induced differential effects on glycolysis, we evaluated the changes in the total ATP production rate and the fractional contribution of the individual pathways to the bioenergetics demands.

In these AML cell lines. We simultaneously measured, in real time, ATP production from the two major key energetic pathways, glycolysis and mitochondrial respiration, using Agilent Seahorse Extracellular Flux analysis. A decrease of approximately 95% in mitochondrial ATP production rates after treatment with 0.1 mM of buformin was measurable in all the three AML cell lines after 12 h. Importantly, the OCI-AML2 and MV4-11 cell lines totally compensated the lower ATP production levels by increasing the glycolytic ATP production (Appendix A).

Conversely, the OCI-AML3 cells could not efficiently switch metabolism and the energetic state of the cells decreased in line with the major sensitivity observed (Appendix A). By adding a high concentration of glucose (4 g/L), the OCI-AML3 cells compensated for the ATP production (Figure 5d), confirming their glycolytic dependence.

The cellular viability, measured by the CellTiter-Glo Assay on the OCI-AML3 cells at low (0.1 g/L) or high (4 g/L) glucose concentrations, increased at high concentrations of glucose, buformin (*p* < 0.0001) and ascorbate–buformin (*p* < 0.0001) (Figure 5e), confirming that the inability of the AML cells to switch to glycolysis is relevant for the induction of apoptosis after ascorbate–buformin treatment.

## 4. Discussion

AML’s phenotypic landscape is characterized by high heterogeneity, clonal evolution and considerable dynamics of genetic and epigenetic events over the course of the disease [1,4,49,50]. Data from clonal evolution studies suggest that the genes commonly involved in epigenetic regulation (i.e., DNMT3A, ASXL1, IDH2 and TET2) are deregulated in pre-leukemic hematopoietic stem cells [2,51]. Such pre-leukemic stem cells are capable of multilineage differentiation, can survive chemotherapy and expand during disease remission. Most of the time, chemotherapy only provides a selective pressure for the expansion of resistant subclones, which vary in their genomic and metabolic phenotypes, and AMLs often relapse [11,50,52,53,54].

To study the efficacy of a metabolic-oriented synergic treatment, we endeavored to define the metabolic landscape and capacity to compensate after the treatment of different AML cells.

We used three cell lines as metabolic models of AMLs. The MV4-11 cells had greater basal OXPHOS values, whereas OCI-AML2 and OCI-AML3 had greater basal glycolysis values. The MV4-11 cells displayed a significant dependency on fatty acid (FA) oxidation fuel and slightly on pyruvate. OCI-AML3 had a strong dependency on glucose fuel and slightly on FAs. OCI-AML2 depended partially on pyruvate and FAs. All these AML cell lines showed high flexibility towards all three fuels. We confirmed the result obtained in the AML cell lines in the primary blasts from AML patients, presenting diverse metabolic backgrounds. Our findings suggest different sensitivities of AML blasts to metabolic therapies, and, in our opinion, acknowledge them as reliable models to foresee the effects of ascorbate–buformin combination therapy in AML patients. Previous studies demonstrated that metformin inhibits the molecular reduction of oxygen in hepatocytes and leukemia cell lines [55,56]. Buformin is chemically related to metformin but is more active and has never been tested in leukemia. Therefore, given its inhibitory effect on complex I, which completely shuts down mitochondrial contributions in ATP production, we addressed its effect on AML cells in combination with ascorbate.

Studies have shown that ascorbate (vitamin C), at pharmacological doses, targets many of the mechanisms that cancer cells utilize for their survival and growth, including redox imbalances and oxygen-sensing regulations [57]. In addition, ascorbate treatment re-establishes TET2 function in AML blasts that present decreased TET2 activity in vitro and in vivo [20,58,59,60]. Altered TET2 function in AMLs can result from heterozygous TET2 mutations and in mutations in IDH1, IDH2 and WT1 [61,62]. Zhao et al. found that ascorbate plus decitabine prior to aclarubicin and cytarabine (A-DCAG) significantly increased the chance of clinical remission after the first induction therapy and extended the median overall survival by 6 months, compared to DCAG alone in AML patients who were over 60 years old [63].

Here, we show that ascorbate inhibits glycolysis by interfering with HK1/2 and GLUT1 functions in hematopoietic cells. The inhibition of HK1/2 and GLUT1 in highly glycolytic OCI-AML3 cells by ascorbate, with the disrupting effect of buformin on the oxygen mitochondrial chain, ultimately leads to an ‘energy crisis’ and cell death. OCI-AML2 and MV4-11 cells, which in basal conditions do not rely upon the glycolysis pathway, switch metabolism and, in part, escape cytotoxicity. Overall, these results suggest that glycolytic behavior is the discriminant between the response and resistance to ascorbate–buformin treatment. This evidence is in line with the higher sensitivity observed in the U937 cells, following the RUNX/RUNX1T1 expression, which turned dependent on glycolysis and were more sensitive to the ascorbate–buformin treatment than the U937-Mock cells. In addition, we observed a positive correlation between the ascorbate treatment and basal glycolytic levels in the six AML samples analyzed. Although it will be necessary to confirm these results in a larger number of AML patient samples, those data clearly depict an effect of ascorbate on glycolysis. Our data contribute to elucidating the targets and mechanisms by which ascorbate and buformin exerts anti-cancer effects. Further insight will be essential for identifying predictive biomarkers for patient stratification and for developing potent combination strategies that lead to durable disease remissions.

Since the efficacy of a treatment should be measured as the capacity of inhibiting the growth of leukemia-initiating cells, by performing clonogenic assays, we demonstrated that an ascorbate–buformin combination treatment induces a drastic reduction in the number of colonies as compared to the untreated AML cell lines and primary blasts from AML patients. The efficiency of the vitamin c treatment in inhibiting leukemic colony formation might also depend on its previously reported ability to restore TET2 function, which drives DNA hypomethylation, by enhancing 5 hmC formations and thereby suppressing leukemic colony formation and the leukemic progression of primary human leukemia patient-derived xenografts (PDXs) [20,60]. Of note that this effect was achieved with low or no toxicity in normal BMC and CD34+ cells from healthy donors, as expected from the higher spare respiratory capacity and high glycolytic reserve of hematopoietic precursors. Overall, our findings indicate that ascorbate–buformin is a suitable therapy to be used, in association with drugs targeting different mechanisms, in fragile patients.

## 5. Conclusions

In conclusion, our data have therapeutic implications especially in fragile patients since both agents have an excellent safety profile, and they support the clinical evaluation of ascorbate–buformin for the treatment of refractory/relapsing AML patients with no other therapeutic options. Since buformin potentiates the effect of ascorbate without adding toxicity, the combination treatment could be associated to other targeted therapies in randomized clinical trials to gauge their utility in the clinic.

## Figures and Tables

**Figure 1 cancers-14-02565-f001:**
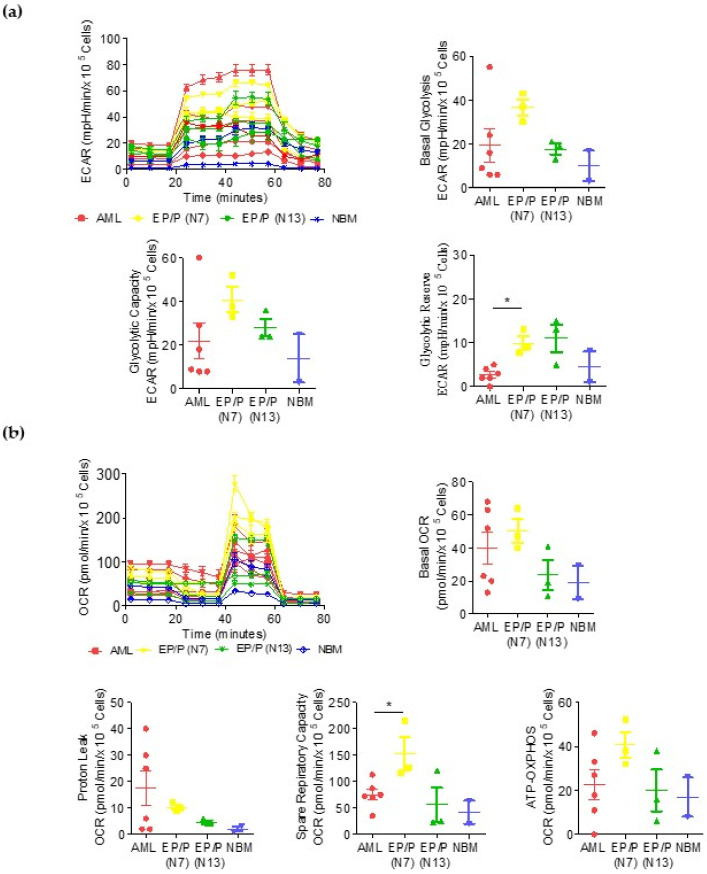
Metabolic characterization of primary AML blasts and early progenitors/precursors (EP/Ps) from cultured normal cord blood CD34+ cells. Metabolic characterization of primary blasts from AML patients, EP/P at day 7 (N7, mostly a promyelocyte population) and at day 13 (N13, mostly granulocytes) and in normal bone marrow (NBM). (**a**) Profile of the glycolytic activity. Histograms represent basal glycolysis, glycolytic reserve and glycolytic capacity measured using the XF Glycolytic Rate Assay. (**b**) Profile of the mitochondrial activity. Histograms represent basal respiration, spare respiratory capacity and mitochondrial ATP production measured using the XF Myto Stress Test Assay. Data are presented as mean ± SD. Statistical analyses were performed using the ANOVA, *t*-test and Tukey’s Multiple Comparison Test; * *p* ≤ 0.05.

**Figure 2 cancers-14-02565-f002:**
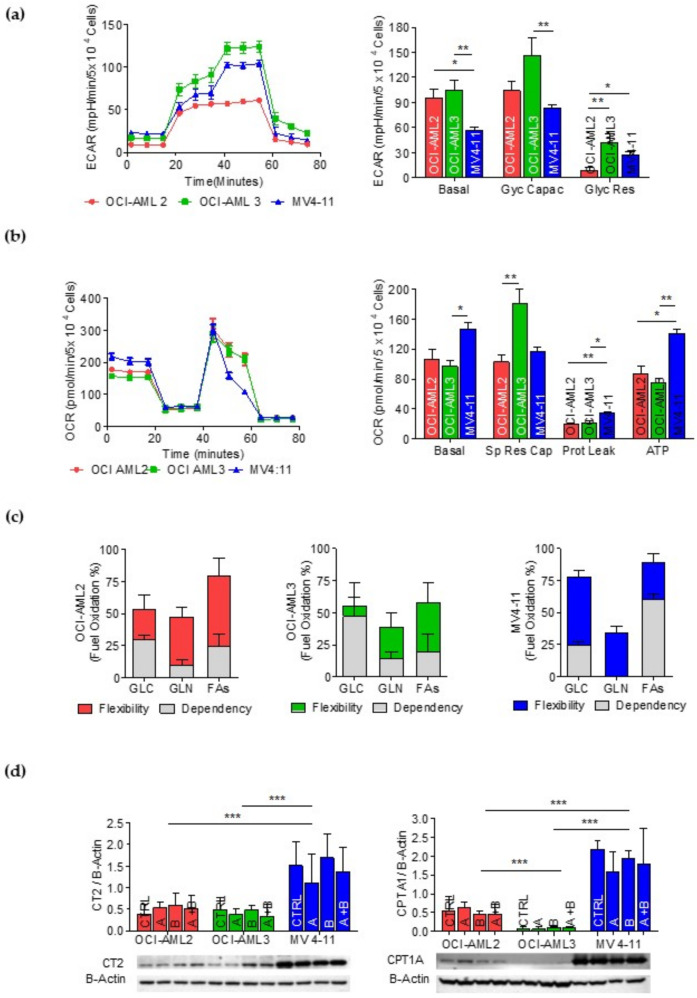
Metabolic characterization of OCI-AML2, OCI-AML3 and MV4-11 cell lines. (**a**) Profile of the glycolytic activity. Histograms represent basal glycolysis, glycolytic reserve and glycolytic capacity. (**b**) Profile of the mitochondrial activity. Histograms represent basal respiration, spare respiratory capacity, proton leak and mitochondrial ATP. (**c**) Evaluation of the mitochondrial fuel used (pyruvate, glutamine and FAs). Data are presented as mean ± SD. The experiments were conducted in triplicate. Statistical significances were evaluated through Kruskal–Wallis one-way ANOVA and Dunn’s post hoc tests. (**d**) CT2 and CPT1A protein expression in OCI-AML2, OCI-AML3 and MV4-11 cell lines treated with 1 mM of ascorbate (A), 0.1 mM of buformin (B) or ascorbate–buformin combination (A + B). Statistical analysis by Student’s *t*-test. * *p* ≤ 0.05; ** *p* < 0.005; *** *p* ≤ 0.0005.

**Figure 3 cancers-14-02565-f003:**
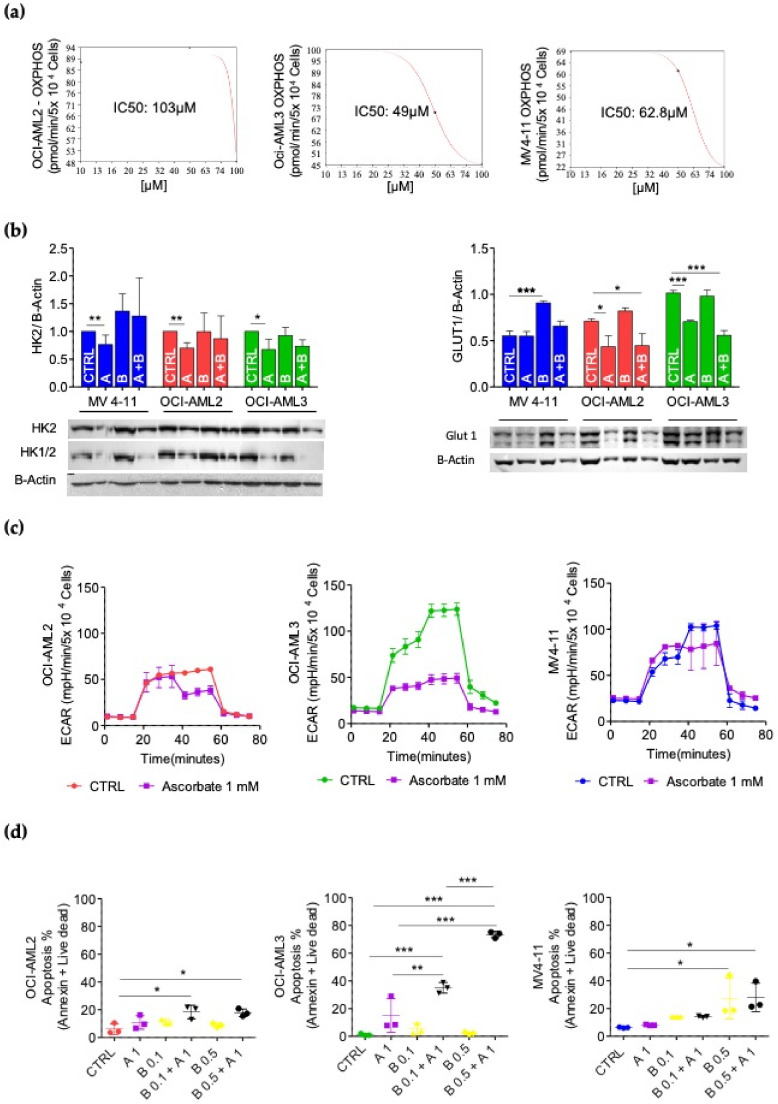
Metabolic effect of buformin and ascorbate in OCI-AML2, OCI-AML3 and MV4-11 cell lines. (**a**) Evaluation of OXPHOS IC50 concentration. Three independent experiments ± SD (**b**) HK2, HK1/2 and GLUT1 protein expression in OCI-AML2, OCI-AML3 and MV4-11 cell lines treated with 1 mM of ascorbate (A), 0.1 mM of buformin (B) or ascorbate–buformin combination (A + B). Statistical analysis by Student’s *t*-test. * *p* ≤ 0.05; ** *p* < 0.005; *** *p* < 0.0005. (**c**) Action of ascorbate on AML cells’ glycolytic metabolism. Kinetic profile of the extracellular acidification rate (ECAR) assay. Cell lines were treated for 24 h with 1 mM of ascorbate and were evaluated by XF Glycolytic Stress Test. The experiments were performed in duplicate. (**d**) Cell death induced by 1 mM of ascorbate (A) and buformin (B) (0.1 and 0.5 mM) at 72 h by flow cytometry after annexin V + live dead staining. Data are presented as mean ± SD from three independent experiments. Statistical analysis by Student’s *t*-test. * *p* < 0.05; ** *p* < 0.005; *** *p* < 0.0005.

**Figure 4 cancers-14-02565-f004:**
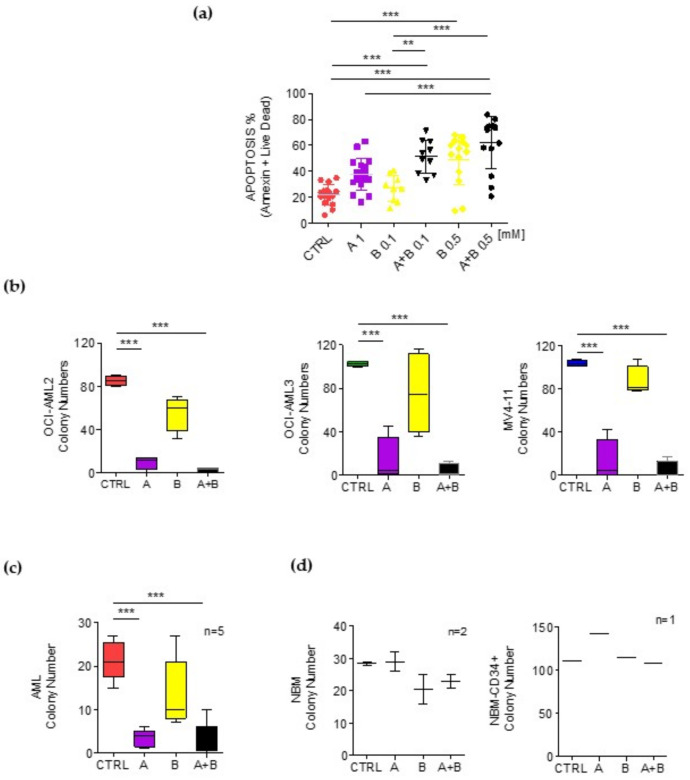
Effects of ascorbate and buformin on survival of AML blasts. Cells were treated with (A) ascorbate (1 mM) and (B) buformin (0.1 and 0.5 mM). (**a**) Cell death of primary blasts from AML patients evaluated by flow cytometry after annexin V + live dead staining. Statistical analysis by Student’s *t*-test. ** *p* ≤ 0.005; *** *p* ≤ 0.0005. (**b**) Clonogenic activity of OCI-AML2, OCI-AML3 and MV4-11 cells treated in semisolid medium for 8 or 13 days. The experiments were conducted in quadruplicate. Statistical analysis by Student’s *t*-test *** *p* ≤ 0.0005. (**c**) Clonogenic activity of AML blasts isolated from the BM of five patients and treated in semisolid medium for 8 days. (**d**) Clonogenic activity of mononucleated cells isolated from the BM of two healthy donors (NBM) (**left**) or CD34+ cells purified from one NBM (**right**) treated in semisolid medium for 8 days. The box plots report the distribution of the number of colony-forming units. Statistical analysis by Student’s *t*-test *** *p* ≤ 0.0005.

**Figure 5 cancers-14-02565-f005:**
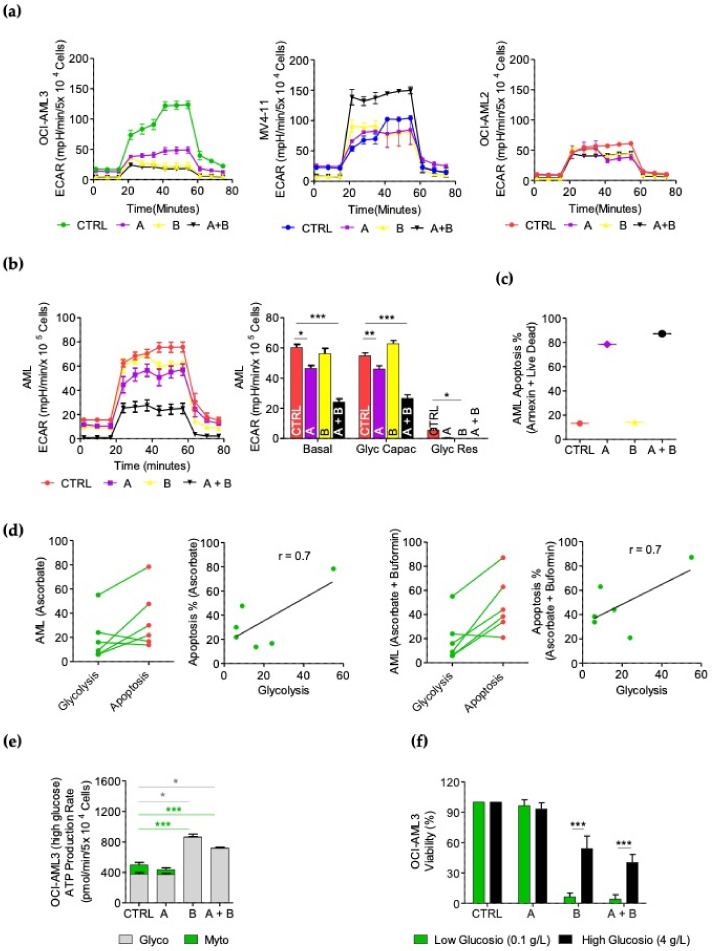
Glycolytic status after buformin plus ascorbate treatment in AMLs. Kinetic profile of ECAR assay in AML cells treated for 24 h with 1 mM of ascorbate (A), 0.1 mM of buformin (B) or ascorbate–buformin combination (A + B) evaluated by XF Glycolytic Stress Test. (**a**) OCI-AML3, MV4-11 and OCI-AML2. (**b**) Primary AML blasts from BM. Histograms represent basal glycolysis, glycolytic capacity (Glyc Capac) and glycolytic reserve (Glyc Res). (**c**) Cell death induced by 1 mM of ascorbate (A), 0.1 mM of buformin (B) or ascorbate–buformin combination (A + B) at 72 h in primary AML blast from the same AML patient by flow cytometry after annexin V + live dead staining. (**d**) Linear correlation between apoptosis and glycolytic levels in AML samples treated with both ascorbate or ascorbate–buformin. (**e**) XF real-time glycolytic and mitochondrial ATP production rate by the ATP Rate Assay in presence of high concentration of glucose (4 g/L) after 12 h of 1 mM of ascorbate (A), 0.1 mM of buformin (B) or ascorbate–buformin combination (A + B) in OCI-AML3 cell line. The experiments were performed in duplicate. (**f**) Cytotoxic efficacy of 1 mM of ascorbate (A), 0.1 mM of buformin (B) or ascorbate–buformin combination (A + B) using the CellTiter-Glo^®^ Luminescent Cell Viability Assay. The cells were cultured at low (0.1 g/L) or high (4 g/L) concentrations of glucose. Three independent experiments were performed in triplicate. Data are presented as mean ± SD. Statistical analysis by Student’s *t*-test. * *p* < 0.05; ** *p* < 0.005; *** *p* < 0.0005.

**Table 1 cancers-14-02565-t001:** Molecular and genetic features of primary AML blasts.

N°	Age	Sex	Molecular Biology	Cytogenetic
* 1	71	F	Negative panel	46, XX
2	64	F	NPM1; FLT3-ITD (R 0.36)	NA
3	79	F	NPM1	FISH negative (chr 5, 7, 8, 11, 20)
4	61	M	Negative panel	46, XY
5	81	F	NPM1; FLT3-ITD (R 0.33)	NA
6	77	M	PLZF/RARa	46, XY, t (11; 17) (q23; q21)
7	76	F	NPM1; FLT3-ITD (R 0.22)	46, XX
8	75	F	FLT3-TKD (AR:0.5)	NA
9	74	M	Negative panel	46, XY
10	51	M	NPM1	FISH negative (chr 5, 7, 8, 11, 20)
11	69	M	Negative panel	46, XY, t (4; 16)
12	60	M	NPM1; FLT3-ITD (R 1.67)	46, XY
13	78	M	NPM1; FLT3-ITD (R 0.58)	NA
14	41	F	MPM1; FLT3-ITD	NA
15	53	F	Negative panel	NA
16	75	F	NPM1; FLT3-ITD (R 0.67)	46, XX
17	45	M	Negative panel	46, XX

NA: not available; Chr: chromosomes; panel of molecular biology (NPM; Nup-Can; FLT3-ITD; FLT3-D835; IDH1; IDH2; CBFb/MYH11; RUNX1/RUNX1T1); * biphenotypic (AML and LLC).

**Table 2 cancers-14-02565-t002:** Glycolysis values in AML blasts and normal hematopoietic cells.

GlycolysisECAR (mpH/min/10^5^ Cells)	AML	EP/P (N7)	EP/P (N13)	NBM	*p*-Value (AML vs. EP/P, N7)
Basal	19 ± 19	37 ± 6	18 ± 4	3 ± 0	-
Capacity	22 ± 20	41 ± 10	28 ± 7	4 ± 1	-
Reserve	3 ± 2	10 ± 3	11 ± 5	0.5 ± 0.7	0.03

ECAR (extracellular acidification rate); EP/P (early progenitors/precursors); N7 (at day 7); N13 (at day 13); NBM (normal bone marrow). Values represent the mean ± SD. Statistical significances were evaluated through Mann–Whitney test.

**Table 3 cancers-14-02565-t003:** Mitochondrial respiration values in AML blasts and normal hematopoietic cells.

Mitochondrial Respiration OCR (pmol/min/10^5^ Cells)	AML	EP/P (N7)	EP/P (N13)	NBM	*p*-Value (AML vs. EP/P)
Basal	40 ± 24	50 ± 12	24 ± 16	9 ± 1	-
Spare Respiratory Capacity	76 ± 25	153 ± 54	56 ± 56	23 ± 4	0.02
Proton Leak	18 ± 16	10 ± 2	5 ± 1	0.5 ± 0.7	-
ATP	23 ± 16	41 ± 10	20 ± 16	8 ± 1	-

OCR (oxygen consume rate); EP/P (early progenitors/precursors); N7 (at day 7); N13 (at day 13); NBM (normal bone marrow). Values represent the mean ± SD. Statistical significances were evaluated through Mann–Whitney test.

**Table 4 cancers-14-02565-t004:** Glycolysis values in AML cell lines.

Glycolysis ECAR (mpH/min/10^5^ Cells)	OCI- AML2	OCI- AML3	MV4-11	*p*-Value	*p*-Value
Basal	96 ± 10	104 ± 12	56 ± 5	<0.5 (Oci2 vs. MV4-11)	<0.005 (OCI3 vs. MV4-11)
Capacity	104 ± 31	146 ± 64	83 ± 10	-	<0.005 (OCI3 vs. MV4-11)
Reserve	9 ± 12	43 ± 32	27 ± 11	<0.005 (OCI2 vs. OCI3)	<0.05 (OCI2 vs. MV4-11)

ECAR (extracellular acidification rate). Values represent the mean ± SD. Statistical significances were evaluated through Kruskal–Wallis one-way ANOVA and Dunn’s post hoc tests.

**Table 5 cancers-14-02565-t005:** Mitochondrial respiration values in AML cell lines.

Mitochondrial Respiration OCR (pmol/min/10^5^ Cells)	OCI- AML2	OCI- AML3	MV4-11	*p*-Value	*p*-Value
Basal	106 ± 31	97 ± 18	160 ± 27	<0.05 MV4-11 vs. OCI3	-
Spare Respiratory Capacity	103 ± 21	180 ± 47	116 ± 15	<0.005 OCI3 vs. OCI2	-
Proton Leak	19 ± 6	22 ± 5	34 ± 4	<0.005 MV4-11 vs. OCI2	<0.05 MV4-11 vs. OCI3
ATP	87 ± 26	75 ± 14	141 ± 13	<0.05 MV4-11 vs. OCI2	<0.005 MV4-11 vs. OCI3

OCR (oxygen consume rate). Values represent the mean ± SD. Statistical significances were evaluated through Kruskal–Wallis one-way ANOVA and Dunn’s post hoc tests.

**Table 6 cancers-14-02565-t006:** Expression levels of CT2 and CPT1A proteins in AML cell lines.

	OCI-AML2	OCI-AML3	MV4-11	*p*-Value (MV4-11 vs. Oci2)	*p*-Value (MV4-11 vs. Oci3)
CT2	0.4 ± 0.2	0.5 ± 0.3	1.5 ± 0.6	<0.0005	<0.0005
CPT1A	0.5 ± 0.4	0.1 ± 0.03	2.2 ± 0.2	<0.0005	<0.0005

Values represent the mean ± SD. Statistical significances were evaluated through Student’s *t*-test.

**Table 7 cancers-14-02565-t007:** Apoptosis effect in AML cell lines and in primary AML blasts treated with ascorbate plus buformin.

	OCI-AML2	OCI-AML3	MV4-11	AML
Ctrl	3 ± 1	3 ± 1	6 ± 1	22 ± 8
Ascorbate 1 mM	6 ± 3	22 ± 13	8 ± 1	38 ± 12
Buformin 0.1 mM	11 ± 2	10 ± 1	14 ± 1	27 ± 10
Ascorbate–buformin 0.1 mM	18 ± 5	33 ± 9	14 ± 1	51 ± 13
Buformin 0.5 mM	7 ± 4	3 ± 2	27 ± 15	49 ± 19
Ascorbate–buformin 0.5 mM	14 ± 8	72 ± 6	28 ± 10	62 ± 20

Values represent the mean ± SD.

**Table 8 cancers-14-02565-t008:** Clonogenic Assay. Colony numbers.

	Ctrl	Ascorbate 1 mM	Buformin (0.1 mM)	Ascorbate–Buformin
OCI-AML2	103 ± 3	11 ± 9	58 ± 16	2 ± 3
OCI-AML3	102 ± 2	14 ± 21	75 ± 38	4 ± 6
MV4-11	104 ± 2	13 ± 20	87 ± 14	4 ± 8
AML blasts	21 ± 5	3 ± 2	14 ± 8	3 ± 4
N-BMC	29 ± 4	29 ± 4	21 ± 6	23 ± 3
CD34+	142	142	115	109

**Table 9 cancers-14-02565-t009:** Cellular Viability response to different metabolic treatments of U937-AETO/U937-Mock cell lines.

	Ctrl	Ascorbate 1 mM	Buformin (0.1 mM)	Ascorbate–Buformin 0.1	Buformin (0.5 mM)	Ascorbate–Buformin 0.5
U937-Mock	1 ± 0	0.81 ± 0.1	0.93 ± 0.06	0.97 ± 0.05	0.85 ± 0.06	0.78 ± 0.1
U937-AETO	1 ± 0	0.67 ± 0.1	0.84 ± 0.1	0.67 ± 0.1	0.80 ± 0.1	0.48 ± 0.01

Values represent the mean ± SD.

## Data Availability

Not applicable.

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
