# Peer review of "Ascorbate Plus Buformin in AML: A Metabolic Targeted Treatment"

_cancers, 2022, doi:10.3390/cancers14102565_

Round 1

Reviewer 1 Report

 The manuscript by Banella et al describes different metabolic activities and responses to drugs targeting metabolism for AML cell lines and primary samples. They first compared various glycolysis and oxphos parameters between AML cells, normal hematopoietic cells at various stages of in vitro differentiation, and normal bone marrow presumably not differentiated in vitro. The purpose of including this experiment is not made clear. They then move on to characterizing various metabolic parameters related to glycolysis and oxphos for AML cell lines and primary samples. They use ascorbate as an inhibitor of glycolysis because it downregulates an enzyme, HK2, involved in glycolysis. However, HK1 also promotes glycolysis, and HK2 has other activities, so the ultimate effect on metabolism is not clear. They use buformin as an inhibitor of oxphos because it blocks ETC complex I. They show that these drugs , alone and in combination, have different effects on AML cells and primary samples, somewhat correlating to the capacity of the AML cells to augment glycolysis when oxphos is blocked. They suggest that these drugs could be studied in frail adults with AML who cannot tolerate traditional chemotherapy. Strengths of the paper are the comprehensive evaluation of various metabolic parameters, and the confirmation of findings in primary samples. General weaknesses are

  1. It is not clear what the differentiation experiments in the beginning add to the story.
  2. It is not clear how exactly ascorbate affects metabolism. They show that it decreases HK2 protein levels. However, HK1 also promotes glycolysis, and HK2 has other activities, so the ultimate effect on metabolism is not clear. Measuring metabolites such as G6P could be useful for deciphering its mechanism.
  3. The relevance of the experiments to in vivo bone marrow is doubtful since influences of various niche stimuli, as well as hypoxia, are not accounted for in these experiments.
  4. Clinical translatability is doubtful since these very high doses still have mostly modest and variable effects.

Other minor comments:

  1. References 30-38 seem to be about metformin, not buformin. Please clarify the text in lines 105-112.
  2. Suggest showing some representative Seahorse curves and explaining how “basal” and “reserve” etc. values are measured the first time these parameters are reported, rather than later in the manuscript.
  3. Line 253: Wouldn’t one expect that reduced spare respiratory capacity would make cells more sensitive to oxidative stress rather than less sensitive? See reference 47.
  4. In Figure 2b the Seahorse curve shows all cell lines with same basal OCR, but the summary bar graph indicates MV4-11 were higher. Please clarify.
  5. For Figure 2c please describe how values for “flexibility” and “dependence” are obtained.
  6. Correct CPTA1 to CPT1A throughout.
  7. Figure 3a is confusing. The text suggests these IC50s are for inhibiting oxphos, though the specific oxphos parameter is not stated. However, the y axes of the dose response curves are labeled “% viability.” What exactly do these IC50s values refer to? If the data refer to viability, they do not match the data in table 7. Please clarify.
  8. Reference 48 cited in line 289 is not about buformin. Please correct.
  9. The conclusion in lines 298-300 that buformin strongly inhibits mitochondrial ATP production is not demonstrated in Figure 3. So far all that has been shown is that high doses are toxic.
  10. It is surprising that ascorbate alone so dramatically reduced colony numbers when it had such a small effect on metabolism and apoptosis in most cases. Can the authors speculate why? Could the effect in the colony assays be due to a different activity of ascorbate, such as those mentioned in lines 499-503?
  11. The authors conclude that sensitivity of AML cells to these metabolically targeted drugs correlates with their dependence on glycolysis. The conclusion might be true but is premature since it is based on one cell line, one transduced cell line and one primary sample. The authors studied 17 patient samples. Can they analyze their data from these samples to determine if there is a correlation between glycolysis dependence and sensitivity to various drug treatments?
  12. How is Figure 5d different from Suppl Figure 1f for OCI-AML3 cells?

Author Response

Dear Rev 1

We wish to thank you for the helpful revision of our manuscript “Ascorbate plus Buformin in AML: a Metabolic Targeted Treatment.”

Detailed responses to the Reviewer’ comments are attached.

Sincerely,

Nelida Ines Noguera

Reviewer 2 Report

In this manuscript, the authors have investigated the metabolic background in different Acute Myeloid Leukemias’ (AMLs) cells, normal hematopoietic progenitors, and patients’ primary AML blasts with Seahorse XF analyzer. They discovered that they have heterogeneous and highly flexible energetic metabolic background. Next, they examined the effect of ascorbate and buformin on these samples, including metabolic rate, cell viability, apoptosis. Overall, they suggested that the combined treatment of ascorbate and buformin would be beneficial to certain AML patients. However, there are some remaining questions to be answered:

1, The authors selected six AML patients to examine the metabolic dependence of their primary AML blasts, could the authors explain why they selected these AML patients from a total of 17 cohort.

2, The authors need to include explanation in Fig 2d figure legends since CTRL, A, B A+B explanation were first shown in Fig 3 instead.

3, Have the authors examine the details on the effects of buformin and ascorbate on AML blasts (Fig 4a) for each individual? The authors have combined all data points and indicated the effect of drugs for the whole population. Considering some patients shared similar molecular biological and cytogenetic background. It is worth to separate them into clusters and investigated the drug effect.

4, Could the authors explain the components of EP/Ps at day 7 (N7)? The percentage of total is over 100%

5, The authors have investigated the drug effect on AML blasts and cells lines with colony formation assay, however, only assessed the drug effect on cell lines with apoptotic assay. Have the authors examined the the drug effect on AML blasts with apoptotic assay as well?

7, The results indicated that the combined treatment only targeted to certain cells. Considering high heterogeneous flexible metabolic background in AML patients as described in this manuscript, could the authors discuss how the treatment would be a therapeutic options for AML patients?

8, typo errors, including:

  • Line 159, “104 cells”
  • Line 190, “PDF membrane”
  • Table 3 and Table 5, “105 cells”
  • Line 269, “grater OXPHOS”
  • Fig 5e, “Glucosio”

Author Response

Dear Rev 2

We wish to thank you for the helpful revision of our manuscript “Ascorbate plus Buformin in AML: a Metabolic Targeted Treatment.”

Detailed responses to the Reviewer’ comments are attached.

Sincerely,

Nelida Ines Noguera

Round 2

Reviewer 1 Report

Thank you for the revisions.